# Trimethylamine N-Oxide Levels Are Associated with Severe Aortic Stenosis and Predict Long-Term Adverse Outcome

**DOI:** 10.3390/jcm12020407

**Published:** 2023-01-04

**Authors:** Yuchao Guo, Shaojun Xu, Hong Zhan, Han Chen, Po Hu, Dao Zhou, Hanyi Dai, Xianbao Liu, Wangxing Hu, Gangjie Zhu, Toru Suzuki, Jian’an Wang

**Affiliations:** 1Department of Cardiology, Second Affiliated Hospital, Zhejiang University School of Medicine, Hangzhou 310009, China; 2Tellgen Corporation, No. 572, Lane 115-1 Bibo Road Zhangjiang Hi-Tech Park Pudong, Shanghai 201203, China; 3Department of Cardiovascular Sciences and NIHR Leicester, Cardiovascular Biomedical Research Centre, University of Leicester, Glenfield Hospital, Leicester LE3 9QP, UK

**Keywords:** trimethylamine N-oxide, aortic stenosis, prognosis, gut microbiota

## Abstract

Objective: Trimethylamine N-oxide (TMAO), a pathological microbial metabolite, is demonstrated to be related to cardiovascular diseases. This study was (1) to investigate the association between TMAO and aortic stenosis and (2) to determine the prognostic value of TMAO for predicting mortality after transcatheter aortic valve replacement (TAVR). Methods: 299 consecutive patients (77 (72–81) years, 58.2% male, Society of Thoracic Surgeons (STS) score 5.8 (4.9–9.3)) with severe aortic stenosis and 711 patients (59 (52–66) years, 51.9% male) without aortic stenosis were included in this retrospective study. A total of 126 pairs of patients were assembled by Propensity Score Matching. The primary outcome was all-cause mortality using survival analyses stratified by TMAO quartiles. Results: Patients with severe aortic stenosis had higher TMAO levels (3.18 (1.77–6.91) μmol/L vs. 1.78 (1.14–2.68) μmol/L, *p* < 0.001), and TMAO remained significantly higher after adjusting for baseline characteristics. Higher TMAO level was associated with higher 2-year all-cause mortality (19.2% vs. 9.5%, log-rank *p* = 0.028) and higher late cumulative mortality (34.2% vs. 19.1%, log-rank *p* = 0.004). In Cox regression multivariate analysis, higher TMAO level remained an independent predictor (hazard ratio 1.788; 95% CI 1.064–3.005, *p* = 0.028) of all-cause mortality after adjusting for STS score, N-terminal pro b-type natriuretic peptide, and maximum velocity. Conclusions: The TMAO level was higher in aortic stenosis patients. Elevated TMAO was associated with poor adverse outcome after TAVR.

## 1. Introduction

Aortic stenosis (AS) is the most common valvular disease around the world [1,2]. Aortic valve replacement is the mainstay of therapy for AS patients. Recently, transcatheter aortic valve replacement (TAVR) has emerged as a safe and effective way for severe AS regardless of risk stratification [1,3]. Some traditional biomarkers (natriuretic peptides, cardiac troponins) or novel ones (growth differentiation factor-15, carbohydrate antigen 125, lipoprotein (a)) are currently being studied for potential utility in early detection of AS and risk stratification after TAVR [4,5,6]. In addition, more translational research studies are warranted to elucidate the underlying AS etiology and reveal new prognostic evaluation tools. Recent studies have linked intestinal microbiota with cardiovascular disease, especially the ways in which dietary nutrients may affect myocardial infarction, heart failure, and hypertension [7]. Trimethylamine N-oxide (TMAO) is a pathological microbial metabolite of dietary choline and phosphatidylcholine found mainly in red meat [8]. Elevated plasma TMAO level has been linked to cardiovascular risk and mortality in recent studies [8,9,10,11]. So far, the role of TMAO in aortic stenosis has not been well studied. A pilot study found higher TMAO and precursor levels in degenerative mitral valve disease in dogs [12]. A gut microbiome study of valvular heart disease and cardiovascular disease by Liu reported that those with valvular heart disease suffered from unique gut microbial dysbiosis [13]. On the other hand, TMAO may promote atherosclerosis, vascular calcification, general inflammation, and platelet activation, which may also contribute to AS. Based on previous studies, we hypothesized that the plasma level of TMAO may be elevated in the AS population and associated with TAVR outcome.

## 2. Methods

### 2.1. Study Population and TAVR Procedures

A total of 299 severe aortic stenosis patients who underwent TAVR and 711 patients with normal coronary angiogram were retrospectively included in this current study from 2013 to 2018 (Appendix A). Aortic stenosis patients were identified from the TORCH registry (NCT02803294). Severe aortic stenosis was diagnosed by echocardiogram criteria (mean gradient ≥40 mmHg, maximum aortic valve velocity ≥4.0 m/s, or aortic valve area ≤1.0 cm^2^). Patients with severe aortic regurgitation or incomplete clinical data were excluded. Most TAVR procedures were completed through transfemoral access. Self-expandable valves including Corevalve (Medtronic Inc., MN, USA), Venus A (Venus Medtech, Hangzhou, China), Vitaflow (Microport, Shanghai, China), TaurusOne (Peijia Medical, Suzhou, China), and their series were used in this study. The balloon expandable valve Sapien XT (Edwards Lifesciences, CA, USA) and the mechanically expandable LOTUS Edge (Boston Scientific Corporation, MA, USA) were also used in this study. All the AS patients were followed up at 1 month, 1 year, and every year thereafter. The primary outcome was defined as all-cause mortality. Analyses for cardiovascular mortality and non-cardiovascular mortality were undertaken as individual secondary endpoints. Survival status was performed by reviewing hospital records or contacting patient family members to ensure no mortality events were missed. 

Patients with normal coronary angiograms from a single center registry at the Second Affiliated Hospital of Zhejiang University were included as the control cohort during the same time period as aortic stenosis patient admission. Exclusion criteria of the control cohort were as follows: any valvular heart disease, cardiomyopathy, arrhythmia, abnormal coronary angiogram (stenosis larger than 50%), incomplete clinical data, and severe comorbidities including chronic or current infections, severe hepatic or renal insufficiency, cancer, or connective tissue diseases. 

This study protocol was approved by the Human Research Ethics Committee at the Second Affiliated Hospital of Zhejiang University (SAHZU) and followed the Declaration of Helsinki. All patients provided written informed consent.

### 2.2. Clinical Characteristics and Laboratory Measurements

Clinical characteristics including age, BMI, gender, smoking history, hyperlipidemia, diabetes mellitus, hypertension, prior stroke history, peripheral vascular diseases (PVD), and chronic obstructive pulmonary disease (COPD) were obtained on admission. Routine echocardiography was performed in all patients, including the AS and control cohorts. Aortic valve type and calcification grade of aortic valve were assessed by multislice computed tomography using 3mensio software (3mensio Medical Imaging BV, Bilthoven, the Netherlands). However, the control cohort did not receive multislice computed tomography. Aortic valve calcifications were graded in a 5-point, semi-quantitative system, as previously reported [14]. Blood samples from the peripheral vein were collected to determine creatinine (Scr), N-terminal portion of brain natriuretic peptide (proBNP), and cardiac troponin (cTn I or T) levels with an automated analyzer (AU5400, Olympus, Tokyo, Japan) [15]. Estimated glomerular filtration rate (eGFR) was calculated with the 2009 Chronic Kidney Disease Epidemiology Collaboration equation. 

### 2.3. TMAO Measurement

Serum samples were collected after diagnosis and before cardiac intervention. Samples were centrifuged, aliquoted, and stored at −80 °C for TMAO measurement. TMAO measurement was performed once for each sample on 7 July 2019. TMAO (96% purity) was purchased from Sigma-Aldrich (Shanghai, China), and its labelled isotope, D9-TMAO (≥98% purity, 99.9% enrichment), was purchased from Cambridge Isotopes (Tewksbury, MA, USA). Sample preparation was performed according to described methods using stable-isotope dilution by mixing 30 μL of plasma with 120 μL of 0.5 μmol/L D9-TMAO in MeOH. Protein precipitation was achieved by a 1 min vortex period followed by centrifugation at 4000× *g* for 5 min. After centrifugation, the supernatant was removed and transferred to a vial for analysis.

The methods of analysis for TMAO were performed as previously described with minor modifications (Tellgen Corporation, Shanghai, China) [16]. Samples were analyzed by liquid chromatography-tandem mass spectrometry with multiple reaction monitoring using an MRM mass analyzer. LC-MS-MRM was performed in positive ion electrospray ionization mode using an LC-30AD UPLC (Shimadzu, Kyoto, Japan). An Acquity UPLC BEH T3 column (130 Å, 1.7 μm, 2.1 mm × 50 mm, Waters Corp., MA, USA) was used. Buffer A was 0.025% NH4OH and 0.045% HCOOH (pH 8.1), and buffer B was pure acetonitrile. An injection volume of 1 μL and a flow of 500 μL/min were used with a column temperature of 50 °C. The gradient started with 4% B at 0 min, returning to 95% B at 0.5 min, maintained to 0.8 min, and dropped to 4% at 1 min, with a total analysis time of 1.0 min. MRM was performed by pre-filtering the precursor ions using the quadrupole mass analyzer for *m*/*z* values of 76.1 (TMAO) and 85.1 (D9-TMAO) and supplying a collision voltage ramp in the transfer cell of 15 to 20 V and 15 to 25 V, respectively. Atomization gas flow was 3 L/min, heating gas flow was 10 L/min, dry gas flow was 10 L/min, interface temperature was 300 °C, DL temperature was 250 °C, and heating block temperature was 400 °C.

### 2.4. Statistical Analysis

Categorical variables are presented as frequencies (proportions) and were compared using χ^2^ tests or Fisher’s exact test when appropriate. Continuous variables are provided as median (interquartile ranges) and were analyzed with nonparametric Mann–Whitney U test for the non-normal distribution of most variables. Logarithmic transformation was used for variables with nonnormal distribution. Propensity score matching (PSM) is used to reduce case-control selection bias and to reduce potential clinical confounders using a 1:1 matching protocol with a caliper width of 0.2. One control was included per AS patient and matched for baseline characteristics including gender, smoker, hyperlipidemia, diabetes, hypertension, prior stroke, PVD, COPD, age, and BMI. Spearman’s correlations analysis was performed between TMAO levels and continuous variables. Logistics regression analysis was performed on the non-propensity matched population to assess the association between TMAO levels, clinical variables, and the presence of aortic stenosis. The Receiver Operating Characteristic (ROC) curve was used to assess the diagnostic ability of TMAO, proBNP, and cTn levels for aortic stenosis. Comparison of area under the ROC curve (AUC) was done by R pack “nsROC” with R ver 3.5.3 [17]. For survival analysis, the Kaplan–Meier methods with log-rank tests were used to depict the time to event curve between TMAO quartile 4 and quartiles 1–3. The hazard ratio (HR) of different variables for late cumulative mortality after TAVR was calculated by a univariate Cox regression. The importance of TMAO as a prognostic factor for late cumulative mortality was investigated in a multivariate Cox analysis model including TMAO, The Society of Thoracic Surgeons’ (STS) risk score, proBNP levels, and maximum velocity. STS risk score was calculated online, based on clinical parameters including creatinine level, age, ejection fraction, bicuspid valve, etc., which were also significant predictors in univariate Cox regression. A two-tailed *p* < 0.05 was considered as statistically significant. All analysis was performed on SPSS version 24 (IBM, Armonk, NY, USA). PSM was performed with PS matching plugin for SPSS [18]. 

## 3. Result

### 3.1. Participants’ Characteristics

The flow chart of patients is presented in Appendix A. Baseline characteristics of 299 AS patients (median age 77 years (IQR 72–81), 58.2% male, median STS score 5.8 (IQR 4.9–9.3)) and 711 normal controls (median age 59 years (IQR 52–66), 51.9% male) are summarized in Table 1, and the AS cohort is further categorized into the above 6.91 mol/L group (quartile 4, N = 73) and the below 6.91 μmol/L group (quartiles 1–3, N = 220). Comparing demographics and clinical variables between AS and control revealed that AS patients were older, had lower BMI, worse cardiac and kidney function and expected lower ejection fraction and higher proBNP and cTn levels. Furthermore, the comorbidities of hyperlipidemia, diabetes, PVD, and COPD were higher in the AS population.

Among 293 AS patients who completed TAVR, 166 (55.5%) patients had more than moderate calcification, and 141 (47.2%) patients were bicuspid valve. With the use of PSM, the demographic difference of the two studied populations was reduced (Appendix A). AS patients with higher TMAO level (≥6.91 mol/L) tend to be older, have lower eGFR and hemoglobin level and higher proBNP, cTn, and STS scores. More patients in the high TMAO group (Q4) were diabetic. No difference was observed between high or low TMAO groups in echocardiographic and procedural variables except pre-dilation rate (100% vs. 94.5%, respectively, *p* = 0.042) (Appendix A).

### 3.2. TMAO Concentrations in Patients with Aortic Stenosis and Control Group

Serum TMAO levels were significantly elevated in patients with AS compared with control (3.18 μmol/L (IQR 1.77–6.91) vs. 1.78 μmol/L (IQR 1.14–2.68), *p* < 0.001) (Table 1, Figure 1B). In PSM-matched AS patients and control cohorts, statistically significant differences of TMAO were also found (*p* = 0.018, Appendix A). In a multiple logistic regression model adjusted for age, smoking history, NYHA class, and ejection fraction, the association between high TMAO and aortic stenosis remained significant (Table 2). ROC curve analysis revealed that TMAO yielded good discrimination ability of AS from control, with an area under the ROC curve (AUC) of 0.69 (95% CI 0.65–073, *p* < 0.001, Figure 1C).

### 3.3. Baseline Clinical Correlates of TMAO in Aortic Stenosis

Furthermore, the correlations between TMAO and other clinical factors were analyzed in patients with AS. No significant association was found between TMAO levels and aortic disease severity parameters such as aortic calcification grades or aortic valve area (Figure 1D,E). Spearman’s correlation analysis revealed that TMAO was positively associated with age, proBNP, and STS score but negatively associated with eGFR and AVA (Appendix A). Further stepwise forward linear regression showed advancing age and worse kidney function (eGFR) as the strongest predictors of TMAO level in the AS population (Appendix A).

### 3.4. TMAO Levels and Post-TAVR Outcome

During a median follow-up of 50 (IQR 36–65) months, 67 patients (22.9%) died in the prognostic TAVR cohort. A higher baseline TMAO (≥6.91 mol/L, quartile 4) was associated with higher late cumulative all-cause mortality, with similar values in cardiac mortality or non-cardiac mortality (Figure 2). Patients with higher baseline TMAO also had a higher 2-year all-cause mortality rate (19.2% vs. 9.5%, log-rank *p* = 0.028, Table 3). However, the 30-day and 1-year mortality rates were not significantly different (*p* = 0.793 and *p* = 0.283, respectively, Table 3). The additional prognostic value of TMAO to STS score or proBNP is shown in Figure 3. The subgroup analysis of TMAO quartiles also revealed a significant increase in late cumulative mortality (*p* for Trend(log-rank) = 0.035, Appendix A).

In univariable Cox analysis, baseline TMAO, proBNP, STS score, eGFR, ejection fraction, valve type (bicuspid valve vs. tricuspid valve), mean gradient, and maximum velocity were independently associated with cumulative mortality (Table 4). In multivariate Cox analysis, after adjusting for STS score, higher TMAO (≥6.91 mol/L, quartile 4) remained an independent predictor for all-cause mortality (HR 1.788; 95% CI 1.064–3.005, *p* = 0.028, Table 4). 

## 4. Discussion

The main finding of our study is that TMAO levels are higher in the AS population and serve as a predictor of all-cause mortality after TAVR.

Previous studies by Suzuki confirm that elevated TMAO is associated with poor prognosis in acute myocardial infarction and acute heart failure, which triggered our research [9,10,11]. Our study does find elevated TMAO levels in the AS population, even adjusting for baseline characteristics. In this study, serum TMAO correlated positively with traditional cardiovascular factors like older age, poor heart and renal function, and high STS score. These cardiovascular risk factors are associated with poor outcome in aortic stenosis [19]. TMAO demonstrates numerous atherosclerosis, thrombosis, and inflammation-promoting effects in recent animal studies, which may share the same pathogenic pathways in aortic stenosis [20]. Our findings were in accordance with a recent paper by Li [21]. They found that TMAO induces osteogenic responses through the endoplasmic reticulum and mitochondrial stress pathways. Aortic valve thicknesses could be reduced by inhibition of TMAO. However, in our present study, TMAO correlated poorly with aortic stenosis severity markers like aortic valve calcification. Our study included more severe aortic stenosis patients compared with previous studies [21]. In our study, aortic valve calcification may be attributed to other etiologies, such as bicuspid valve and rheumatic heart disease.

In the present study, we demonstrated for the first time that higher TMAO levels were associated with adverse cardiac and non-cardiac outcome after TAVR. These results are consistent with the prognosis effect investigation of TMAO in heart failure, diabetes mellitus, myocardial infarction, and chronic kidney disease. As these are all common comorbidities in aortic stenosis patients [7]. TMAO may contribute to platelet hypersensitivity and thrombus development, which may result in TAVR prothesis valve thrombosis and stroke [22]. High TMAO may also be a marker of impaired renal function because TMAO is excreted mainly by the kidneys [7,23]. In turn, TMAO elevation also aggravates renal function [24]. It is widely accepted that poor renal function is an adverse prognostic factor after TAVR [25]. In our study, renal dysfunction is associated with elevated TMAO in liner regression and is also associated with poor outcome in the univariate Cox regression. Lastly, our observations of the connection between TMAO and non-cardiac death, mostly cancer (31.6% of total death) in our cohort, are in accordance with studies that link TMAO to various cancers [26]. In our present study, TMAO served as an independent predictor even adjusted by STS score, which may be instrumental in residual risk identification after TAVR.

### Limitations

The major limitation of this study is its relatively small sample size and lack of validation cohort. In particular, the size of the AS population in our single center was limited. On the other hand, a multicenter validation TAVR cohort will eliminate demographic variation. Comparisons with healthy community volunteers, different degrees of AS patients, and the AS cohort without intervention will provide more information about the discrimination ability of TMAO. Secondly, the analyses of the intestinal microflora structure and amount of red meat consumption were not included in this study, which will be confirmed in future studies.

## 5. Conclusions

This work provides a comprehensive assessment of TMAO in an aortic stenosis population. TMAO levels were elevated in severe aortic stenosis patients. TMAO was associated with poor renal function and cardiovascular risk factors in the AS population. Elevated TMAO seems associated with poor adverse outcomes in post-TAVR follow-up and provides additional prognostic information.

## Figures and Tables

**Figure 1 jcm-12-00407-f001:**
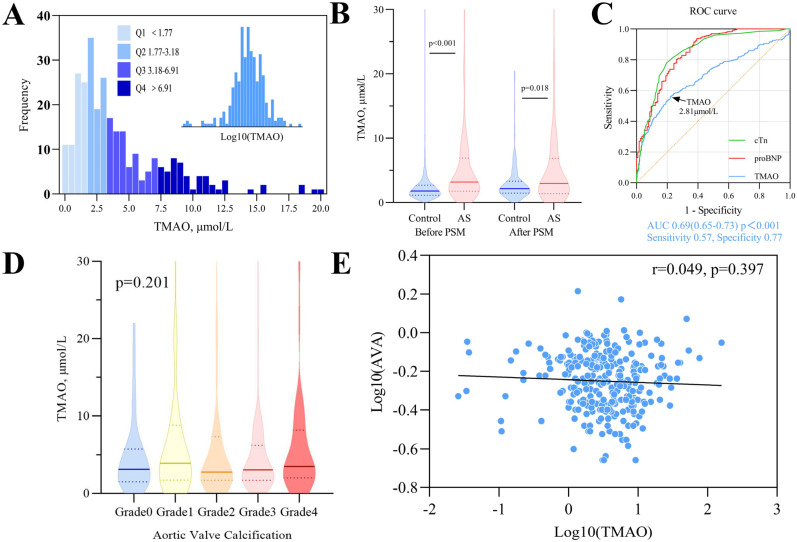
Serum TMAO levels in patients with or without aortic stenosis. (**A**) Frequency distribution histogram of serum TMAO levels in AS population, n = 299. (**B**) Violin plot of TMAO levels in patients with AS (n = 299) or without AS (control group, n = 711) before and after propensity matching (n = 126 in both groups). (**C**) TMAO achieved a receiver operating characteristic curve area under the curve of 0.69 with a best cut-off value of 2.81μmol/L. (**D**) Violin plot showing no significant difference of TMAO levels in different grades of aortic valve calcification (one-way ANOVA *p* = 0.201). Data are presented as violin plots with median (solid line) and quartiles (dotted line) in (**B**,**D**). (**E**) Correlation between TMAO and aortic valve area (AVA) in AS after common logarithm transformation. TMAO = Trimethylamine N-oxide; AS = aortic stenosis; cTn = cardiac troponin; proBNP = N-terminal prohormone of brain natriuretic peptide; AVA = aortic valve area.

**Figure 2 jcm-12-00407-f002:**
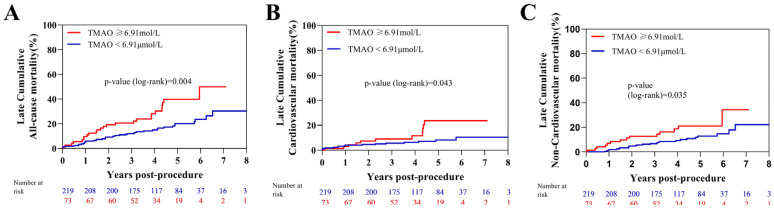
Late cumulative and 2-year event rate of mortality according to elevated TMAO. Kaplan–Meier analysis of late cumulative all-cause mortality (**A**), cardiac mortality (**B**), and non-cardiac mortality (**C**) stratified by TMAO levels in AS patient after TAVR. TAVR = transcatheter aortic valve replacement. TMAO = Trimethylamine N-oxide.

**Figure 3 jcm-12-00407-f003:**
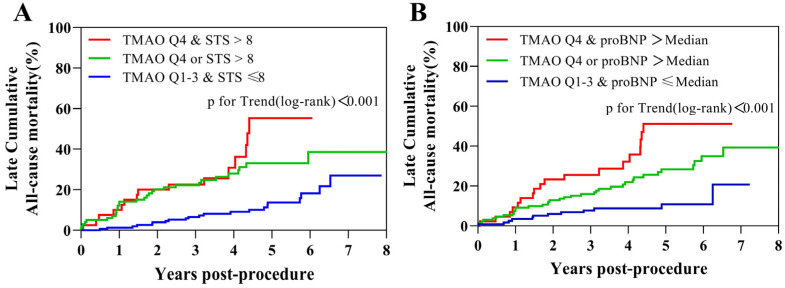
Late cumulative all-cause mortality according to combinations of TMAO and STS or proBNP. Kaplan–Meier survival analysis for late cumulative all-cause mortality stratified by (**A**) combinations of TMAO and STS score and (**B**) combinations of TMAO and proBNP, reporting none, one, or both elevated. TMAO above 6.91 μmol/L (Q4), proBNP above 2871 pg/mL (median), STS score above 8 (high risk), respectively. TMAO = Trimethylamine N-oxide; STS = Society of Thoracic Surgeons; proBNP = N-terminal prohormone of brain natriuretic peptide.

**Table 1 jcm-12-00407-t001:** Characteristics of aortic stenosis patients and controls.

	Controls	AS	AS with TMAO	
	N = 711	N = 299	<6.91 μmol/L N = 220	≥6.91 mol/L N = 73
**Clinical Variables**				
Age, years	59 (52–66)	77 (72–81) *	76.5 (72–80.75)	80 (75–83) †
Male (%)	369 (51.9)	174 (58.2)	125 (56.8)	45 (61.6)
BMI, kg/m^2^	24.2 (22.1–26.5)	22.8 (20.0–24.8) *	22.9 (20.0–24.8)	22.6 (20.0–25.0)
NYHA III/IV (%)	43 (7.4)	266 (89.0) *	195 (88.6)	66 (90.4)
Smoker (%)	251 (35.3)	38 (12.7) *	26 (11.8)	11 (15.1)
Hyperlipidemia (%)	86 (12.1)	68 (22.7) *	54 (24.5)	13 (17.8)
Diabetes (%)	85 (12.0)	66 (22.1) *	42 (19.1)	22 (30.1)
Hypertension (%)	345 (48.5)	163 (54.5)	113 (51.4)	46 (63)
Prior stroke (%)	37 (5.2)	17 (5.7)	11 (5)	6 (8.2)
PVD (%)	69 (9.7)	66 (22.1) *	45 (20.5)	20 (27.4)
COPD (%)	36 (5.1)	68 (22.7) *	47 (21.4)	20 (27.4)
eGFR, mL/(min × 1.73 m^2^)	99 (87–108)	75 (55–89) *	79 (60–91)	60 (33–79) *
**Biomarkers**				
proBNP, pg/mL	123 (56–871)	2945 (987–8863) *	2598 (787–7595)	3932 (1542–11,776) ‡
cTn, ng/mL	0.008 (0.005–0.013)	0.0275 (0.017–0.050) *	0.03 (0.01–0.05)	0.03 (0.02–0.06) ‡
TMAO, μmol/L	1.78 (1.14–2.68)	3.18 (1.77–6.91) *	2.40 (1.37–2.67)	10.42 (8.58–19.06) *
**Hemodynamics**				
EF, %	65.9 (60.1–70.5)	57.3 (44.2–64.4) *	57.5 (44.55–64.7)	57.2 (43.95–62.05)
AVA, cm^2^	-	0.58 (0.44–0.72)	0.59 (0.44–0.73)	0.55 (0.44–0.68)
Mean gradient, mmHg	-	53 (42–67)	53 (42–67)	54 (43–63)
Maximum velocity, m/s	-	4.8 (4.3–5.3)	4.76 (4.2–5.33)	4.82 (4.39–5.22)
**AS- and TAVR-related Variables**				
STS score	-	5.8 (4.0–9.3)	5.3 (3.7–8.9)	8.6 (4.9–12.0)
Bicuspid valve	-	141 (47.2)	107 (48.6)	30 (41.1)
Calcification > moderate	-	166 (55.5)	123 (55.9)	39 (53.4)
Transfemoral route	-	279 (95.2)	208 (94.5)	71 (97.3)
Prothesis type				
BEV	-	25 (8.5)	17 (7.7)	8 (11)
SEV	-	246 (84)	187 (85)	59 (80.8)
Lotus	-	22 (7.5)	16 (7.3)	6 (8.2)
Pre-dilation (%)	-	281 (95.9)	208 (94.5)	73 (100)
Post-dilation (%)	-	133 (45.4)	106 (48.2)	27 (37) ‡
Second valve used (%)	-	22 (7.5)	18 (8.2)	4 (5.5)

Values are n (%) or median (interquartile range). Bold: statistically significant differences. Comparison between AS and controls or AS with TMAO ≥ 6.91 mol/L group (Quartiles 4, Q4) and <6.91 μmol/L group (quartiles 1–3, Q1–3), * *p* < 0.001, † *p* < 0.01, ‡ *p* < 0.05. AS = aortic stenosis; TMAO = Trimethylamine N-oxide; cTn = cardiac troponin; AVA = aortic valve area; BMI = body mass index; NYHA = New York Heart Association; PVD = peripheral vascular diseases; COPD = chronic obstructive pulmonary disease; eGFR = estimated glomerular filtration rate; EF = ejection fraction; STS = Society of Thoracic Surgeons; BEV = balloon expandable valve; SEV = self-expending valve.

**Table 2 jcm-12-00407-t002:** Association between clinical characteristics and aortic stenosis by logistics regression.

	Unadjusted Model			Adjusted Model		
	OR	95% CI	*p*	OR	95% CI	*p*
Age, years	1.267	1.229–1.305	**<0.001**	1.230	1.172–1.291	**<0.001**
BMI, kg/m^2^	0.851	0.815–0.889	**<0.001**	-	-	-
NYHA class	17.994	11.794–24.516	**<0.001**	20.298	11.598–35.523	**<0.001**
Male (%)	1.290	0.982–1.695	0.067	-	-	-
Smoker (%)	0.267	0.184–0.388	**<0.001**	0.427	0.19–0.962	**0.040**
Hyperlipidemia (%)	2.139	1.504–3.042	**<0.001**	-	-	-
Diabetes (%)	2.086	1.463–2.975	**<0.001**	-	-	-
Hypertension (%)	1.271	0.97–1.667	0.082	-	-	-
PVD (%)	2.636	1.822–3.813	**<0.001**	-	-	-
COPD (%)	5.519	3.588–8.491	**<0.001**	-	-	-
eGFR, mL/(min × 1.73 m^2^)	0.940	0.931–0.948	**<0.001**	-	-	-
EF, %	0.942	0.931–0.953	**<0.001**	1.036	1.007–1.067	**0.015**
TMAO, μmol/L	1.217	1.157–1.279	**<0.001**	1.084	1.01–1.162	**0.025**

In unadjusted model, only correlations for which *p* < 0.1 were shown. A multiple logistic regression model with forward likelihood selection was used to adjust covariates. Bold: statistically significant differences. OR = odds ratio; CI = confidence interval; BMI = body mass index; NYHA = New York Heart Association; PVD = peripheral vascular diseases; COPD = chronic obstructive pulmonary disease; eGFR = estimated glomerular filtration rate; EF = ejection fraction; TMAO = Trimethylamine N-oxide.

**Table 3 jcm-12-00407-t003:** Clinical outcome of patients after TAVR according to quartiles of TMAO (Q4 vs. Q1–3).

Clinical Outcome	AS Patients	AS with TMAO		*p*
	N = 293	<6.91 mol/L N = 220	≥6.91 mol/L N = 73	
Follow up duration (months)	50 (36–65)	50 (37–66)	44 (35–60)	**0.045**
30-day mortality (%)	5 (1.7)	4 (1.8)	1 (1.4)	0.793
Cardiac	4 (1.4)	4 (1.8)	0 (0)	0.248
Non-cardiac	1 (0.3)	0 (0)	1 (1.4)	0.085
1-year mortality (%)	20 (6.8)	13 (5.9)	7 (9.6)	0.283
Cardiac	15 (5.1)	11 (5)	4 (5.5)	0.858
Non-cardiac	5 (1.7)	2 (0.9)	3 (4.1)	0.067
2-year mortality (%)	35 (11.9)	21 (9.5)	14 (19.2)	**0.028**
Cardiac	16 (5.5)	11 (5)	5 (6.8)	0.520
Non-cardiac	19 (6.5)	10 (4.5)	9 (12.3)	**0.017**
Late cumulative mortality (%)	67 (22.9)	42 (19.1)	25 (34.2)	**0.004**
Cardiac	28 (9.6)	17 (7.7)	11 (15.1)	**0.043**
Non-cardiac	39 (13.3)	25 (11.4)	14 (19.2)	**0.035**

Kaplan–Meier estimates of mortality between AS patient with higher TMAO (≥6.91 mol/L, Q4) and low TMAO (<6.91μmol/L, Q1–3). Comparisons were made with the log-rank test. Bold: statistically significant differences. AS = aortic stenosis; TMAO = Trimethylamine N-oxide.

**Table 4 jcm-12-00407-t004:** Association with late cumulative all-cause mortality after TAVR (Cox regression analysis).

	Univariate Analysis *		Multivariate Analysis **	
	Hazard Ratio (95% CI)	*p*-Value	Hazard Ratio (95% CI)	*p*-Value
TMAO ≥ 6.91 mol/L (Q4 vs. Q1–3)	2.065 (1.255–3.396)	**0.004**	1.788(1.064–3.005)	**0.028**
STS score > 8	2.675 (1.645–4.352)	**<0.001**	1.837(1.091–3.093)	**0.022**
eGFR < 30 mL/(min × 1.73 m^2^)	4.728 (2.466–9.066)	**<0.001**		
EF < 50%	1.793 (1.107–2.903)	**0.018**		
Bicuspid valve	0.491 (0.29–0.829)	**0.008**		
proBNP > 2871 pg/mL (median)	2.629 (1.544–4.476)	**<0.001**	2.141(1.230–3.726)	**0.007**
Mean gradient > 53 mmHg (median)	0.559 (0.342–0.916)	**0.021**		
Maximum velocity > 4.78 m/s(median)	0.557 (0.34–0.912)	**0.020**	0.556(0.337–0.918)	**0.022**

* In univariate analysis, only variants with *p* < 0.05 were shown. ** Renal function (eGFR or creatine), EF, and bicuspid valve type were not selected in multivariate model, as they were included in STS score. Mean gradient had high linear correlation with maximum velocity, which was also omitted. Bold: statistically significant differences. CI = confidence interval; TMAO = Trimethylamine N-oxide; STS = Society of Thoracic Surgeons; eGFR = estimated glomerular filtration rate; EF = ejection fraction; proBNP = N-terminal prohormone of brain natriuretic peptide.

## Data Availability

The data presented in this study are available in the article and Appendix A.

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
