# Peer review of "Trimethylamine N-Oxide Levels Are Associated with Severe Aortic Stenosis and Predict Long-Term Adverse Outcome"

_jcm, 2023, doi:10.3390/jcm12020407_

Round 1
Reviewer 1 Report
The present paper is an original article looking at a relatively new and promsising biomarker, i.e. Trimethylamine N-Oxide (TMAO) in patients with aortic stenosis (AS) and controls. The authors studied 299 patients with severe AS undergoing transcatheter aortic valve replacement and 711 controls without AS. Propensity score matching was applied, and after matching there were were 126 pairs. Patients with severe AS had higher TMAO than subjects without AS, and this was also true after propensity score matching. In addition, higher TMAO was associated with higher two year mortality among patients with AS.
General comment: this an interesting study as it helps to further characterize this biomarker. In particular, the prognostic data are relevant. It would be interesting what the mechanism of higher TMAO in those with poor prognosis is. TMAO will not be used as a tool the diagnose AS as the AUC is not good enough for a diagnostic test, and a blood biomarker can never replace an echocardiogram. The authors also showed that NT-proBNP is a better predictor of AS than TMAO. However, the parameter is interesting in terms of risk prediction. Is there any information hemodynamics and how TMAO was related to hemodynamics.
Specific comments:
TMAO analysis was performed based on frozen samples. Were all plasma samples analysed at the same time, and is it known whether TMAO is stable in frozen samples?
TMAO was used as a categorical variable (tertiles). Was the prognostic effect also seen when used as a continuous variable?
Is there any information hemodynamics and how TMAO was related to hemodynamics.
Author Response
We sincerely thank the editor and all reviewers for their valuable feedback that we have used to improve the quality of our manuscript. The reviewer comments are laid out below and specific concerns have been numbered. Our response is given in normal font and changes/additions to the manuscript are given in red text.
Point 1: General comment: this an interesting study as it helps to further characterize this biomarker. In particular, the prognostic data are relevant. It would be interesting what the mechanism of higher TMAO in those with poor prognosis is. TMAO will not be used as a tool the diagnose AS as the AUC is not good enough for a diagnostic test, and a blood biomarker can never replace an echocardiogram. The authors also showed that NT-proBNP is a better predictor of AS than TMAO. However, the parameter is interesting in terms of risk prediction. Is there any information hemodynamics and how TMAO was related to hemodynamics.
Response 1: As mentioned in Table S2, no correlations were found between TMAO and hemodynamics variables (EF, Mean gradient, AVA, Max velocity, PASP) in Aortic Stenosis patients. We found a negative correlation between AVA and TMAO, as shown in Table S2. However, AVA and TMAO data required log10 transformation to assure normality. The correlation between AVA and TMAO was lost after log10 transformation.
Point 2: TMAO analysis was performed based on frozen samples. Were all plasma samples analysed at the same time, and is it known whether TMAO is stable in frozen samples?
Response 2: Yes, all plasma samples were analyzed at the same time. According to published literature (DOI: 10.1016/j.ab.2014.03.016), “Stability studies reveal that TMAO in plasma is stable both during storage at -80°C for 5 years and to multiple freeze thaw cycles.” Our samples were storage at -80°C and never melted before analysis.
Point 3: TMAO was used as a categorical variable (tertiles). Was the prognostic effect also seen when used as a continuous variable?
Response 3: As mentioned in Table S4, the prognostic effect of TMAO (continuous variable, per 1 mol/L) was also observed with HR 1.015(1.005-1.026), P = 0.002 for all-cause mortality.
Point 4: Is there any information hemodynamics and how TMAO was related to hemodynamics.
Response 4: Same as response 1.

Reviewer 2 Report
This is an original topic that would be of interest to the readers of the journal. It is generally well structured. The manuscript is clear and straight to the point. I have provided few minor revisions below.
Abstract:
§ In the results section:
§ and remained significant different: what does the authors mean?
§ Higher TMAO level was associate with: associated
§ higher TMAO level remained a independent: an independent
§ Kindly mention the full name for the acronyms, such as, STS, BNP, TAVR
Page 8: mean gradient and max velocity were independently associated
§ Re-mention abbreviations in each footnote
In discussion:
§ Li et al.
§ The authors mentioned: “Our study included more severe aortic stenosis patients regardless
§ compared with previous one”, what are the references for the previous one?
§ “It is widely accepted that poor renal function is an adverse prognostic factor after TAVR”: missing reference
§ “In our study renal function is associated with elevated TMAO in liner regression and also poor outcome in univariate Cox regression.”: merit rephrasing
Author Response
We sincerely thank the editor and all reviewers for their valuable feedback that we have used to improve the quality of our manuscript. The reviewer comments are laid out below and specific concerns have been numbered. Our response is given in normal font and changes/additions to the manuscript are given in red text.
We tried our best to improve the manuscript and made some changes in the manuscript. These changes will not influence the content and framework of the paper. And here we did not list the changes but marked in red in revised paper. We appreciate for Reviewers’ warm work earnestly and hope that the correction will meet with approval.
Point 1: § In the results section:
and remained significant different: what does the authors mean?
Higher TMAO level was associate with: associated
higher TMAO level remained a independent: an independent
Kindly mention the full name for the acronyms, such as, STS, BNP, TAVR
Response 1: We sincerely thank the reviewer for careful reading. As suggested by the reviewer, we have corrected the “and remained significant different” into “and TMAO remained significant higher”. We have carefully checked the manuscript and corrected the errors accordingly.
Point 2: Page 8: mean gradient and max velocity were independently associated
Re-mention abbreviations in each footnote
Response 2: We have revised each footnote.
Point3: In discussion:
Li et al.
The authors mentioned: “Our study included more severe aortic stenosis patients regardless
compared with previous one”, what are the references for the previous one?
Response3:We sincerely appreciate the valuable comments. “previous one ” refers to reference [21]. We have add reference [21] after this sentence.
- Li J, Zeng Q, Xiong Z, Xian G, Liu Z, Zhan Q, Lai W, Ao L, Meng X, Ren H, Xu D. Trimethylamine -N-oxide induces osteogenic responses in human aortic valve interstitial cells in vitro and aggravates aortic valve lesions in mice. Cardiovasc Res 2021.
Point4 § “It is widely accepted that poor renal function is an adverse prognostic factor after TAVR”: missing reference
Response4 : As suggested by the reviewer, we have added more references to support this idea (26. Gupta, T.; Goel, K.; Kolte, D.; Khera, S.; Villablanca, P.A.; Aronow, W.S.; Bortnick, A.E.; Slovut, D.P.; Taub, C.C.; Kizer, J.R.; et al. Association of chronic kidney disease with In-Hospital outcomes of transcatheter aortic valve replacement. JACC Cardio-vasc Interv 2017, 10, 2050-2060.).
Point 5 § “In our study renal function is associated with elevated TMAO in liner regression and also poor outcome in univariate Cox regression.”: merit rephrasing
Response 5: We have rephrased this sentence. “In our study, renal dysfunction is associated with elevated TMAO in liner regression, and also associated with poor outcome in univariate Cox regression.”
